# The nuclear lamina couples mechanical forces to cell fate in the preimplantation embryo via actin organization

Robin M. Skory [1,2,6], Adam A. Moverley[1,3,6], Goli Ardestani [4], Yanina Alvarez[5], Ana Domingo-Muelas [1], Oz Pomp[1], Blake Hernandez[1], Piotr Tetlak[1], Stephanie Bissiere[1], Claudio D. Stern [3], Denny Sakkas [4] ✉ & Nicolas Plachta [1] ✉

During preimplantation development, contractile forces generated at the apical cortex segregate cells into inner and outer positions of the embryo, establishing the inner cell mass (ICM) and trophectoderm. To which extent these forces influence ICM-trophectoderm fate remains unresolved. Here, we found that the nuclear lamina is coupled to the cortex via an F-actin meshwork in mouse and human embryos. Actomyosin contractility increases during development, upregulating Lamin-A levels, but upon internalization cells lose their apical cortex and downregulate Lamin-A. Low Lamin-A shifts the localization of actin nucleators from nucleus to cytoplasm increasing cytoplasmic F-actin abundance. This results in stabilization of Amot, Yap phosphorylation and acquisition of ICM over trophectoderm fate. By contrast, in outer cells, Lamin-A levels increase with contractility. This prevents Yap phosphorylation enabling Cdx2 to specify the trophectoderm. Thus, forces transmitted to the nuclear lamina control actin organization to differentially regulate the factors specifying lineage identity.

Mechanical forces shape most embryonic tissues and organs, yet it remains a challenge to reveal how cells sense these forces and adapt their fate in response to them in vivo. During mammalian development, the first lineage segregation occurs when the cells of the preimplantation embryo are physically sorted into the pluripotent inner cell mass (ICM) or outer trophectoderm to form the fetus and placenta, respectively[1]. This segregation starts during the 8- to 16-cell stage in mouse embryos. Following division, some cells are directly allocated to an inner or outer position. Yet in most cases, mechanical forces generated by the actomyosin cortex drive cell internalization via apical constriction, a process whereby cells gradually reduce their apical cortex area until they become fully internalized within the inner mass[2].

Specification of ICM or trophectoderm fate in inner and outer cells is controlled via differential regulation of key transcription factors. Outer cells express high Cdx2 levels and adopt a trophectoderm identity[3,4]. Cdx2 expression requires nuclear localization of the transcriptional co-activator Yap[5], which occurs when Yap is unphosphorylated. Current models propose that in inner cells, the Hippo pathway member Angiomotin (Amot) localizes to the basolateral cortex, where it promotes Yap phosphorylation and its retention within the cytoplasm. By contrast, the polarized F-actin–rich apical cortex of

[1]Department of Cell and Developmental Biology, Institute for Regenerative Medicine Perelman School of Medicine, University of Pennsylvania, Philadelphia, PA, USA. [2]Department of Obstetrics and Gynecology, Division of Reproductive Endocrinology and Infertility, Perelman School of Medicine, University of Pennsylvania, Philadelphia, PA, USA. [3]University College London, WC1E 6BT London, UK. [4]Boston IVF-Eugin Group, Waltham, MA, USA. [5]Universidad de Ciencias Exactas y Naturales, Universidad de Buenos Aires, Buenos Aires, Argentina. [6]These authors contributed equally: Robin M. Skory, Adam A. Moverley. ✉e-mail: dsakkas@bostonivf.com; nicolas.plachta@pennmedicine.upenn.edu

outer cells is thought to sequester Amot from Yap, allowing Yap nuclear localization[6–9].

Some studies proposed that the polarized apical domain of 8-cell stage blastomeres could be asymmetrically inherited during cell division to differentially regulate Yap[10–12]. However, live-embryo imaging revealed that these apical domains are largely disassembled before cell division, instead of being asymmetrically inherited[8,13]. Moreover, the timing of polarization of the apical cortex after division displays substantial variability between outer cells[14]. Thus, it is unclear to which extent apical polarity regulates Yap during early stages of inner-outer cell segregation. In addition to models based on cell polarity, work in cell cultures has identified various mechanical stimuli affecting Yap localization[15–18]. Thus, ICM-trophectoderm fate could also be regulated by differences in actomyosin contractility preceding the establishment of differences in polarity, yet the mechanistic basis for this is unknown.

The nuclear lamina is well poised to link mechanical forces and cell fate. It is composed of Lamin intermediate filaments, connected to the cytoskeleton via LINC proteins[19]. A-type Lamins (Lamin-A/C) are alternatively spliced products of the *LMNA* gene, and B-type Lamins are encoded by *LMNB1* and *LMNB2*. *LMNA* knockout mice complete preimplantation development, but Lamin-A functions have been proposed to be compensated by multiple proteins[20] including truncated forms of Lamin[21]. These knockouts also display cardiac dysfunction, associated with defective mechanosensing[22,23]. Interestingly, biochemical measurements showed that Lamin-A levels are markedly different between soft tissues like brain, and stiffer ones like bone[24]. Lamin-A levels also respond to mechanical stimuli in culture conditions[24,25] and can be modulated via cell differentiation[26,27]. Furthermore, human laminopathies such as Emery–Dreifuss muscular dystrophy and dilated cardiomyopathy are associated with mutations in *LMNA*, and other genes linked to nuclear mechanosensing such as Emerin and Nesprin[28,29]. Despite this progress in linking Lamins to mechanosensing processes, the roles of the nuclear lamina in responding to forces or regulating cell fate during mammalian development remain elusive.

Here, we explore the role of the nuclear lamina during the first lineage segregation event in preimplantation development. Lamin-A is mechanically coupled to the cell cortex and responds to forces driving inner-outer segregation. These changes in Lamin-A trigger actin reorganization and downstream effects on Yap and Cdx2, key factors specifying ICM and trophectoderm fate.

## Results

### Lamin-A levels scale with contractility in the early embryo

To investigate the roles of the nuclear lamina during preimplantation development we first analyzed Lamin-A localization and levels in the early embryo, prior to inner-outer cell segregation. Mechanical stimulation can promote Lamin-A localization to the nuclear periphery, where it forms a major component of the nuclear lamina[30]. However, Lamin-A phosphorylation causes disassembly from the lamina, nucleoplasm localization and increased turnover[24,25,31]. Thus, Lamin-A response to mechanical forces can be investigated by measuring changes in its lamina versus nucleoplasm ratio. To explore this, we performed immunofluorescence for endogenous Lamin-A/C in the intact embryo, followed by computational segmentation of the lamina and nucleoplasm subnuclear compartments. We then quantified the ratio between lamina and nucleoplasm fluorescence intensity, hereafter referred to as Lamin-A/C L:N (Supplementary Fig. 1a). This reveals a progressive increase in Lamin-A/C L:N between the 2-cell stage and the compacted 8-cell stage (Fig. 1a, b). By contrast, total Lamin-A/C levels do not increase, but display a minor decrease over time (Supplementary Fig. 1b), indicating that changes in Lamin-A during these developmental stages are primarily characterized by a shift in subnuclear distribution, rather than total nuclear levels.

Mechanical forces have been shown to regulate Lamin-A levels in other systems[24,25], and actomyosin contractility in particular plays important roles in controlling cell shape and inner-outer cell segregation during preimplantation development[32]. Thus, we investigated whether the increase in Lamin-A/C L:N is regulated by contractility in the embryo. Analysis of nuclear morphology revealed that the nuclear lamina displays a more wrinkled morphology at earlier stages, characterized by more distortion, elongation, invaginations and bleb-like structures, but becomes more taught at later developmental stages (Supplementary Fig. 1c–e), consistent with rising contractility[24,25]. Furthermore, immunofluorescence shows that the levels of phosphorylated myosin II increase from the 2- to 8-cell stage (Fig. 1c, d). During cell division, blastomeres reduce their cell-cell contact surface, and their shape during cytokinesis is largely determined by cortical tension[33,34]. Live-embryo imaging demonstrates that at the 4-cell stage, dividing blastomeres display a dynamic and highly irregular morphology during cytokinesis (Fig. 1f; Supplementary Movie 1). In contrast, 8-cell blastomeres maintain a more spherical shape during division, in line with higher contractile forces. In addition to changes in cell shape, embryo compaction has been associated with rising contractility, as 8-cell blastomeres increase their apical surface area[35,36]. Correspondingly, Lamin-A/C L:N increases after compaction (Fig. 1a, b). To further probe the link between actomyosin contractility and Lamin-A L:N, we treated embryos with the ROCK inhibitor H-1152. This causes a reduction in Lamin-A/C L:N and triggers the adoption of less spherical lamina morphologies (Fig. 1a, e; Supplementary Fig. 1f). In contrast, treatment with H-1152 at the two-cell stage when contractility is lower has little effect on Lamin-A/C L:N or nuclear morphology (Supplementary Fig. 1g). Together, these results suggest that rising contractility during early preimplantation development not only drives changes in nuclear morphology and cell shape, but also leads to an increase in Lamin-A/C L:N.

### Contractile forces control Lamin-A via the F-actin meshwork

The mammalian cell nucleus can be linked to the cortex via the three main components of the cytoskeleton, comprising microtubules, actin and intermediate filaments[37–39], yet it is unknown how these networks are organized between the cortex and nucleus in the preimplantation embryo[32]. To explore their role, we microinjected siRNAs into one cell of the 2-cell embryo for SUN1/2, which couples the nuclear lamina to the cytoskeleton[19]. SUN1/2 knockdown causes a downregulation of Lamin-A/C L:N (Supplementary Fig. 2a), which raises the question of whether Lamin-A levels in the embryo could be regulated by interactions with the cytoskeleton. Thus, we imaged the organization of each cytoskeletal component. Immunofluorescence for α-tubulin reveals an enrichment of microtubules at perinuclear regions (Supplementary Fig. 2b), yet treatment with the microtubule depolymerizing drug nocodazole does not disrupt Lamin-A L:N (Supplementary Fig. 2c), indicating that microtubules are not the main cytoskeletal component regulating Lamin-A L:N in the embryo. Keratins 8 and 18 (K8 and K18) establish a filamentous network under the apical cortex and manipulating their levels affects Yap and Cdx2[8]. However, while keratin filaments are present at the 16-cell stage, they do not form dense networks until the blastocyst stage (Supplementary Fig. 2d). Furthermore, keratin filaments at the 16-cell stage remain confined to the cortex and thus are unlikely to interact with the nuclear lamina directly at this stage. Consistent with this, microinjection of siRNAs targeting K8 and K18 disrupts the keratin network of the embryo[8] but does affect inner outer differences in Lamin-A L:N (Supplementary Fig. 2e).

F-actin is enriched at the cell cortex throughout preimplantation development. However, we previously showed that it also forms a meshwork throughout the cytoplasm[8]. Analysis of this meshwork from the 2- to 8-cell stage demonstrates that it spans the space between the cortex and nuclear lamina in mouse (Fig. 1g) and shows a similar distribution in human embryos (Fig. 1h). Moreover, our analysis of

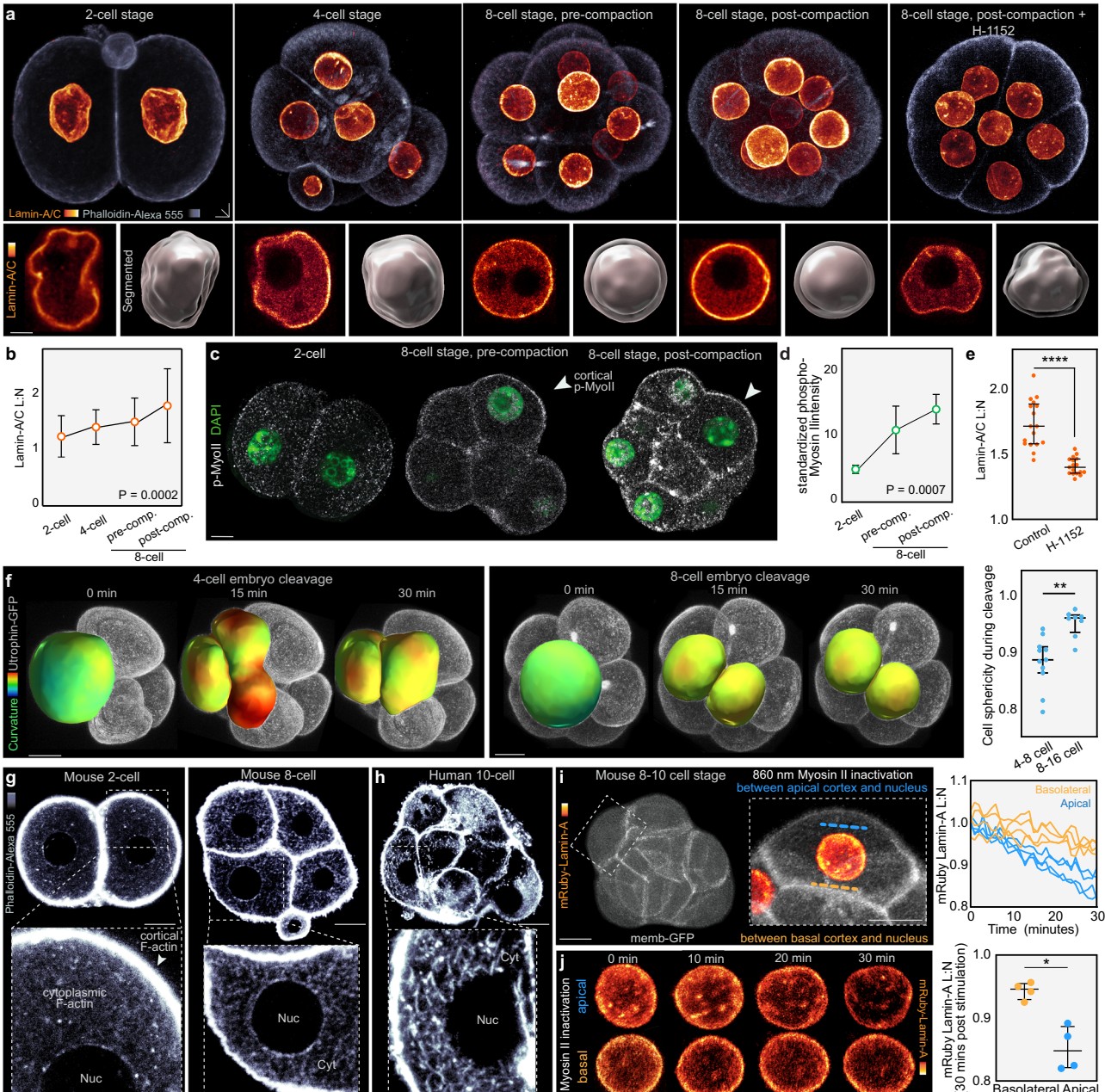

**Fig. 1 | Lamin-A L:N responds to changes in actomyosin contractility prior to lineage segregation. a, b** 3D immunofluorescence and quantification of Lamin-A/C lamina:nucleoplasm levels (Lamin-A/C L:N) in intact mouse embryos shows that Lamin-A/C L:N increases during development prior to lineage segregation. This is accompanied by changes in nuclear shape, with nuclei becoming more spherical over time. Treatment with H-1152 (50 μM, 4 h) causes a reduction in Lamin-A/C L:N and nuclear sphericity. Insets show 1 μm sections of the nucleus and 3D segmentation of the lamina (transparent) and nucleoplasm (opaque). Dots represent the mean and error bars represent SD ($n = 10$ for 2-cell, $n = 34$ for 4-cell, $n = 59$ for pre-compaction 8-cell, $n = 54$ for post-compaction 8-cell; $P = 0.0002$, Kruskal-Wallis test). **c, d** Immunofluorescence and quantification (fluorescence intensity standardized to DAPI) shows rising phosphorylated-myosin II between the 2-, 8-, and 16-cell stages. Dots represent the mean and error bars represent SD. $n = 4$ for 2-cell, $n = 14$ for 8 cell, and $n = 16$ for 16-cell. $P = 0.0007$ Kruskal-Wallis test. **e** Lamin-A/C L:N decreases after treatment with H-1152 at the 8-cell stage. $n = 17$ for control and H-1152. ****$P < 0.0001$, Mann-Whitney U test. **f,** Live-imaging of 4-cell and 8-cell

cleavage divisions reveals greater deformation of cell shape at early stages. Utrophin-GFP allows 3D segmentation and analysis of local curvature. $n = 11$ for 4-8 cell, $n = 8$ for 8-16 cell. **$P = 0.006$, Mann-Whitney U test. Staining with phalloidin reveals an F-actin meshwork throughout the cytoplasm in early-stage mouse (**g**) and human (**h**) embryos. The meshwork extends from the cell cortex to the nucleus and the cytoplasmic density remains similar between the 2- and 8-cell stages. Representative examples selected from phalloidin stainings of >100 mouse embryos (**g**) and 9 human embryos (**h**). **i, j** Disrupting the contractility of the F-actin meshwork causes a reduction in mRuby-Lamin-A L:N. Spatiotemporally controlled stimulation of azido-blebbistatin was performed using a 860 nm laser targeting the F-actin meshwork between the nucleus and cortex (**i**). Graph shows mRuby-Lamin-A L:N following photostimulation. $n = 4$ for apical and basolateral, $p = 0.0286$, Mann-Whitney U test. All statistical tests are two-tailed. Bars in dot plots represent median and interquartile range. Scale bars, 10 μm. Source data are provided as a Source Data file.

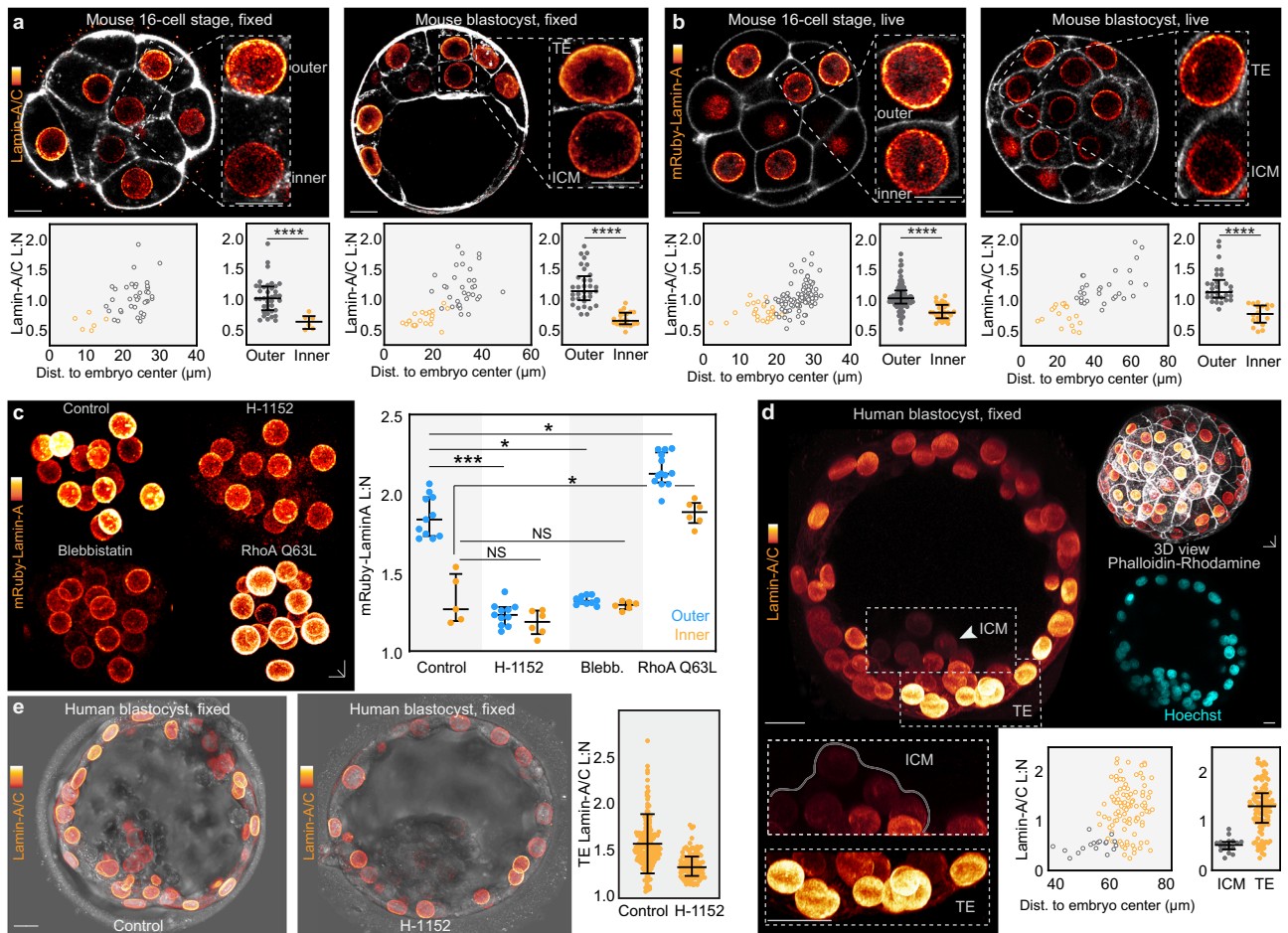

**Fig. 2 | Differences in Lamin-A L:N identify lineage segregation in mouse and human embryos. a, b,** Immunofluorescence for Lamin-A/C in fixed mouse embryos (**a**) and live-imaging of mRuby-Lamin-A (**b**) at different developmental stages. Images show thin confocal sections to illustrate inner-outer differences. Lamin-A L:N plotted against distance to embryo center of mass. For (**a**) $n = 6$ for 16-cell inner, $n = 40$ for 16-cell outer, $n = 21$ for blastocyst ICM, $n = 33$ for blastocyst trophectoderm. For (**b**) $n = 29$ for 16-cell inner, $n = 48$ for 16-cell outer, $n = 18$ for blastocyst ICM, $n = 32$ for blastocyst trophectoderm. ****$P < 0.0001$, Mann-Whitney U test. **c,** Live-imaging and quantification of mRuby-Lamin-A in embryos treated with H-1152 or blebbistatin show reduced mRuby-Lamin-A L:N. Conversely, embryos expressing the constitutively active RhoA mutant Q63L show increased mRuby-Lamin-A L:N. $n = 11$ for control outer, $n = 5$ for control inner, $n = 11$ for H-1152 outer, $n = 6$ for H-1152 inner, $n = 10$ for blebbistatin outer, $n = 6$ for blebbistatin

inner, $n = 13$ for RhoA Q63L outer and $n = 6$ for RhoA Q63L inner. ***$P < 0.0007$, *$P = 0.0343$ for control outer vs blebbistatin outer, *$P = 0.0410$ for control outer vs RhoA Q63L outer. *$P = 0.0418$ control inner vs RhoA Q63L inner. Mann-Whitney U test. **d,** Human blastocyst immunostained for Lamin-A/C shows lower Lamin-A L:N in the ICM versus trophectoderm (TE). Quantification shows the relationship between Lamin-A/C L:N and distance to embryo center of mass. $n = 19$ for ICM, $n = 108$ for trophectoderm (two human blastocysts were analyzed and therefore statistical analysis was not performed). **e,** Human blastocysts treated with ROCK inhibitor H-1152 have reduced Lamin-A/C L:N (one human blastocyst per group, trophectoderm cells $n = 199$ for control, $n = 95$ for H-1152). All statistical tests are two-tailed. Bars in dot plots represent median and interquartile range. Scale bars, 10 μm. Source data are provided as a Source Data file.

phosphorylated myosin II shows that in addition to the cortex, it also localizes throughout the cytoplasm (Fig. 1c). This suggests that cortical and cytoplasmic actomyosin networks may provide a continuous contractile system transmitting forces to the cell nucleus. To test this, we applied azido-blebbistatin, which allows spatiotemporal control of myosin II inhibition via selective illumination of subcellular regions with 860 nm light[40]. Targeting the 860 nm laser between the nucleus and cortex causes a progressive reduction in mRuby-Lamin-A L:N (Fig. 1i, j). This effect is more pronounced when the stimulation is performed at the apical versus basal region, consistent with higher levels of phospho-myosin II at the apical region[2,11]. Thus, these results indicate that actomyosin contractility controls Lamin-A L:N via the F-actin meshwork.

**Lamin-A levels identify the first lineage segregation**
As our results indicate that Lamin-A responds to changes in actomyosin contractility in vivo we next examined its levels during inner-

outer lineage segregation, at the 8- to 16-cell stage. This revealed that inner cells display lower Lamin-A/C L:N compared to outer cells, confirmed by two Lamin-A/C antibodies (Fig. 2a; Supplementary Fig. 3a). Furthermore, this inner-outer difference is maintained at the blastocyst stage between the cells of the ICM and trophectoderm (>32-cells) (Fig. 2a). Unlike Lamin-A/C L:N, inner and outer cells show similar levels of Lamin-B1 L:N (Supplementary Fig. 3b), consistent with overall normal nuclear mechanics found in Lamin-B1–deficient cells[24,41–43].

To supplement our analysis by immunostaining, we microinjected embryos with mRNA for Lamin-A fused to mRuby (mRuby-Lamin-A) and imaged its dynamics during inner-outer cell segregation. Similar to endogenous Lamin-A/C probed by immunofluorescence (Fig. 2b) and to previous work using Lamins tagged with fluorescent proteins[24,31], mRuby-Lamin-A localizes to the nucleus and is enriched at the lamina during interphase, with a smaller fraction in the nucleoplasm (Fig. 2b). Furthermore, inner cells display lower mRuby-Lamin-A L:N than outer cells, consistent with the endogenous Lamin-A/C L:N pattern (Fig. 2b).

Most available antibodies recognize both the Lamin-A and C isoforms, and thus we cannot distinguish how each isoform responds to changes in cell position. However, our experiments imaging mRuby-Lamin-A alone indicate that this isoform behaves similarly to endogenous Lamin-A/C analyzed via immunofluorescence (Fig. 2a). Moreover, immunofluorescence using a Lamin-A/C Ser22 phospho-specific antibody, which is thought to identify the non-filamentous component confined to the nucleoplasm, demonstrates higher levels in the inner cell nucleoplasm compared to outer cells[31] (Supplementary Fig. 3c). Consistent with this, fluorescence recovery after photobleaching (FRAP) experiments demonstrate a larger immobile fraction of mRuby-Lamin-A in outer than inner cells (Supplementary Fig. 3d).

Performing cell segmentation shows that as cells internalize, they reduce their aspect ratio, flatness and apical:total surface area (Supplementary Fig. 3e). In addition, mRuby-Lamin-A L:N show the highest correlation ($R^2$) with apical/total surface area (Supplementary Fig. 3e). This is in line with our demonstration that the area of the apical cortex where the level of myosin II and actomyosin contractility is higher decreases during cell internalization[2]. Thus, to further probe the link between Lamin-A L:N and actomyosin contractility during inner-outer segregation, we treated 8- to 16-cell embryos with H-1152 and blebbistatin, which causes all nuclei within the embryo to display lower mRuby-Lamin-A L:N (Fig. 2c). Conversely, we expressed the constitutively-active RhoA mutant Q63L that increases actomyosin contractility[44,45], which resulted in an increase in mRuby-Lamin-A L:N throughout the embryo (Fig. 2c).

We then explored whether the nuclear lamina is also regulated by actomyosin contractility in human preimplantation embryos. Given the limited availability of vitrified early human embryos (8-cell to morula), we focused on the blastocyst stage. In line with our analyses in the mouse embryo, immunofluorescence in the human blastocyst reveals a similar trend to mouse blastocysts, with lower Lamin-A/C L:N in the ICM, compared to trophectoderm cells (Fig. 2d). Moreover, H-1152 treatment causes a decrease in Lamin-A/C L:N in the human blastocyst (Fig. 2e). These results suggest that Lamin-A L:N scale up during development due to the action of actomyosin contractility, but is downregulated when cells constrict their apical cortex and internalize to form the ICM.

## Differences in Lamin-A control Yap and Cdx2

To explore whether changes in Lamin-A L:N during inner-outer cell segregation correspond to the expression pattern of cell fate markers, we first used immunofluorescence for endogenous Lamin-A/C and phospho-Yap, which reveals an inverse relationship between their levels (Fig. 3a). Outer cells display the highest Lamin-A/C L:N and lowest levels of cytoplasmic phospho-Yap. In contrast, cells showing features of apical constriction and fully internalized cells display lower Lamin-A/C L:N and higher cytoplasmic phospho-Yap (Fig. 3a). Moreover, immunofluorescence for Lamin-A/C and Cdx2 shows that outer cells displaying the highest Lamin-A/C L:N also show the highest Cdx2 levels (Fig. 3a).

To further explore this, we imaged mRuby-Lamin-A and Yap tagged with GFP (Yap-GFP) in live embryos. To avoid unwanted effects resulting from Yap overexpression we microinjected a low Yap-GFP mRNA concentration. In order to simultaneously track cell position we also co-injected a membrane-GFP mRNA, which we have previously demonstrated does not contribute fluorescence to the nucleus, and is restricted to the membrane[46]. This permitted examination of Yap-GFP nuclear-cytoplasmic ratios in vivo and revealed that the exclusion of Yap-GFP from the cell nucleus correlates with both, apical constriction and a decrease in mRuby-Lamin-A L:N (Supplementary Fig. 4a).

*LMNA* gene knockout mice complete preimplantation development[22,23], but display compensation by multiple proteins[20,21]. Thus, we employed a knockdown strategy that reduces, but does not completely eliminate Lamin-A/C expression by microinjecting siRNAs

targeting Lamin-A/C. We performed microinjection either into the zygote or into one cell of the 2-cell embryo, with the latter approach enabling comparison of knockdown and control cells within the same embryo (Supplementary Figs. 4b,c). In outer cells, Lamin-A knockdown causes an increase in the cytoplasmic levels of phospho-Yap, and a reduction in nuclear Cdx2 (Fig. 3b,c). Moreover, staining for total Yap shows a reduction in nuclear levels in knockdown outer cells (Fig. 3d). Therefore, Lamin-A not only responds to the forces that segregate inner and outer cells, but also controls the downstream transcriptional regulators of cell fate Yap and Cdx2.

## The actin meshwork is regulated by Lamin-A and stabilizes Amot

To test how Lamin-A controls Yap and Cdx2, we performed immunofluorescence for the Yap regulator Amot. Current models propose that the polarized apical cortex found in outer cells sequesters Amot allowing Yap nuclear localization, while in inner cells Amot localizes to the basolateral cortex where it promotes Yap phosphorylation and its cytoplasmic retention[6–9]. Using an Amot antibody against the C-terminus recognizing both p130 and p80 isoforms[7], we confirmed that Amot localizes to both the apical cortex of outer cells and basolateral cortex of inner cells (Fig. 4a). However, these experiments also identify a sharp difference in the cytoplasmic abundance of Amot, with inner cells displaying markedly higher levels than outer cells (Fig. 4a).

To quantitatively analyze these differences, we used tools for computational segmentation of the cortex and cytoplasm (Supplementary Fig. 4d). After characterizing the endogenous levels of Amot within inner and outer cells, we tested whether our Lamin-A knockdown disrupts this pattern. Embryos injected with Lamin-A siRNA showed increased Amot levels in the cytoplasm of outer cells (Fig. 4b), suggesting that they adopt a phenotype similar to that of inner cells. Analyses of human blastocysts demonstrates that Amot levels are also higher throughout the cytoplasm in the ICM, compared to the trophectoderm (Fig. 4f). Therefore, our results confirm previous studies proposing that a fraction of Amot localizes to the cell cortex. However, we also identify prominent differences in Amot levels throughout the cytoplasm of mouse and human embryos distinguishing inner and outer cells. Notably, the extensive Yap phosphorylation found in the cytoplasm (Fig. 3a) is consistent with the high levels of Amot in this location (Fig. 4a, f).

Amot can bind to actin in biochemical assays[47] and we found an extensive F-actin meshwork throughout the cytoplasm in the embryo (Fig. 1g, h). Importantly, analysis of this meshwork during inner-outer segregation reveals that it becomes more dense in inner, compared to outer cells (Fig. 4c). In line with these results, FRAP experiments show a larger immobile fraction of fluorescently-tagged Amot (Emerald-Amot) in the inner cell cytoplasm, compared to outer cells (Supplementary Fig. 4e). To further examine whether this dense F-actin network is responsible for stabilizing Amot in inner cells we disrupted the meshwork using Latrunculin A, resulting in lower levels of cytoplasmic Amot in inner cells (Fig. 4d). Moreover, this F-actin pattern is similarly observed in human blastocysts, which display a denser cytoplasmic F-actin meshwork within the ICM, compared to the trophectoderm (Fig. 4g). Work in cell culture demonstrates that Lamin-A can regulate actin dynamics[48]. Hence, we hypothesized that Lamin-A may also regulate the density of the F-actin meshwork in the embryo. Consistent with this, Lamin-A knockdown causes an accumulation of F-actin in the cytoplasm of outer cells, which adopt a similar pattern to that found in inner cells (Fig. 4e). Thus, our results indicate that Lamin-A regulates actin organization in the embryo, and identifies the cytoplasmic actin meshwork as an extensive scaffold for Amot stabilization.

In addition to regulating fate markers through actin organization, it is plausible that changes in Lamin-A/C could impact chromatin reorganization. To explore this, we examined sub-nuclear chromatin organization using DAPI and immunostaining for H3K9me3. This revealed no changes in heterochromatin levels at the nuclear

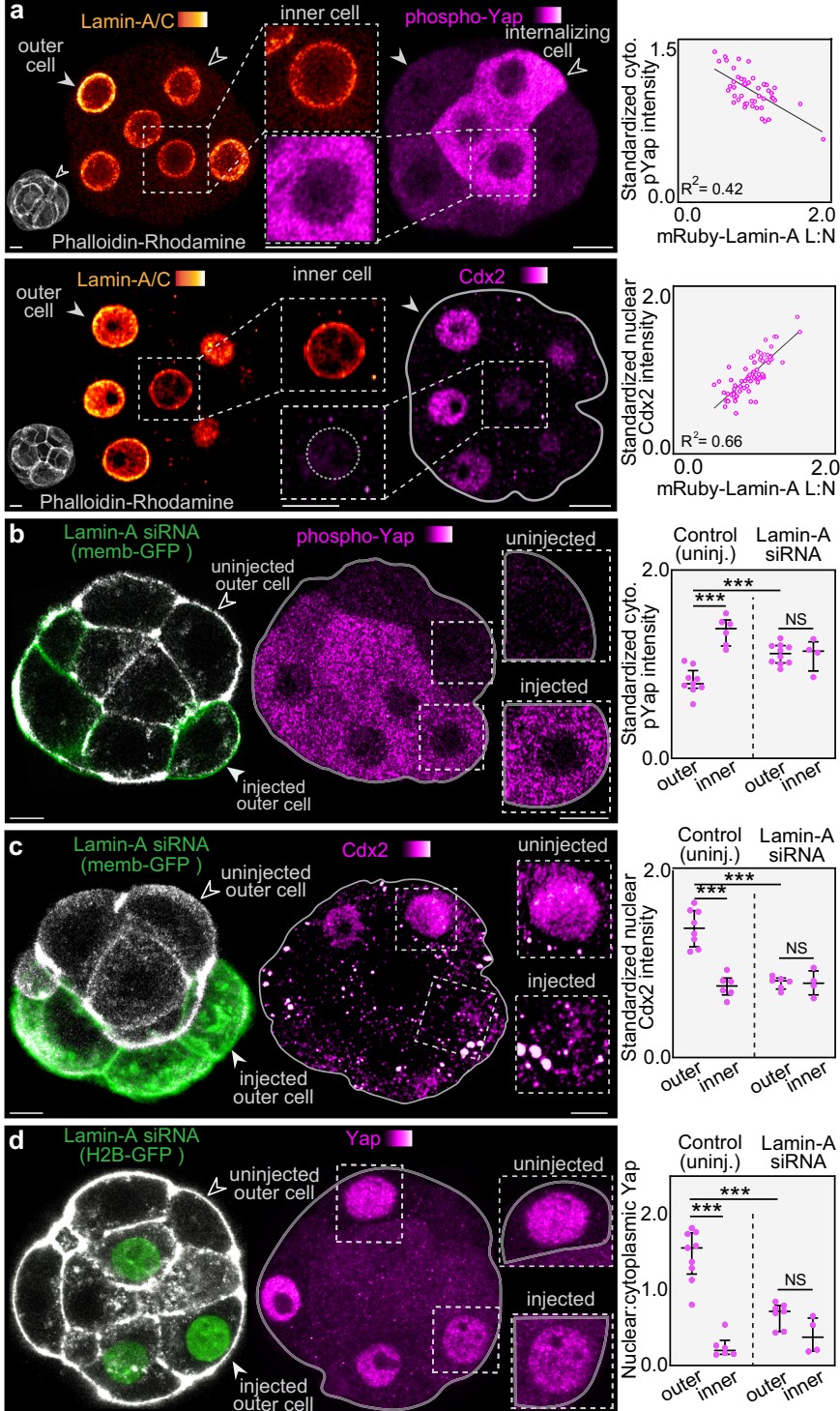

**Fig. 3 | Lamin-A controls the transcriptional regulators Yap and Cdx2.**
**a**, Cytoplasmic intensity of phospho-Yap and nuclear Cdx2 correlate with Lamin-A/
C L:N. Open arrowhead indicates a cell undergoing internalization via apical con-
striction. $n = 48$ for phospho-Yap, $n = 71$ for Cdx2, $R^2 = 0.42$ for phospho-Yap,
$R^2 = 0.66$ for Cdx2. **b**–**d**, Knockdown of Lamin-A by siRNA causes an increase in
cytoplasmic phospho-Yap levels (**b**), a reduction in nuclear Cdx2 levels (**c**) and a
reduction in nuclear:cytoplasmic Yap levels (**d**). siRNAs were microinjected into a
single cell of the 2-cell embryo. Membrane-GFP or H2B-GFP was used to label
injected cells (arrowheads). For (**b**) $n = 9$ for control outer, $n = 6$ for control inner,
$n = 10$ for Lamin-A siRNA outer, $n = 4$ for Lamin-A siRNA inner. ***P = 0.0004 for

control inner vs control outer, ***P = 0.0006 for control outer vs Lamin-A siRNA
outer, NS > 0.999. For (**c**) n = 8 for control outer, $n = 6$ control inner, $n = 6$ Lamin-A
siRNA outer, $n = 4$ Lamin-A siRNA inner. ***P = 0.0007 for control inner vs control
outer, ***P = 0.0007 for control outer vs Lamin-A siRNA outer, NS = 0.9143. For (**d**)
$n = 9$ for control outer, $n = 6$ for control inner, $n = 7$ for Lamin-A siRNA outer, $n = 4$
for Lamin-A siRNA inner. ***P = 0.0004 for control inner vs control outer,
***P = 0.0003 for control outer vs Lamin-A siRNA outer, NS = 0.0727. All statistical
tests are two-tailed. Fluorescence intensities represent results standardized to
DAPI. Bars in dot plots represent median and interquartile range. Scale bars, 10 μm
Source data are provided as a Source Data file.

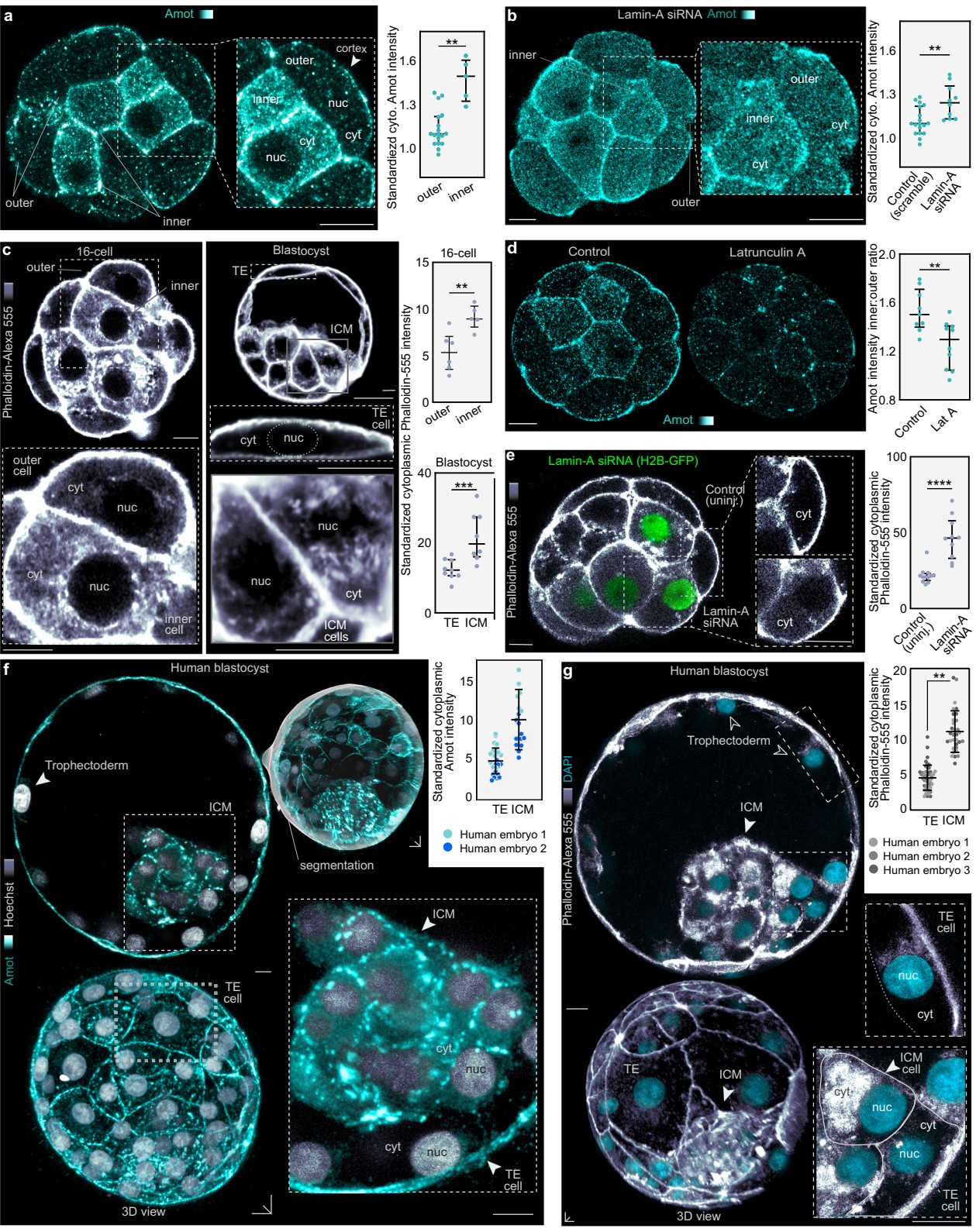

periphery of inner and outer cells, or following Lamin-A knockdown (Supplementary Fig. 5), which suggest no broad changes in sub-nuclear chromatin organization in response to changes in Lamin-A/C L:N at these stages.

## Lamin-A regulates Formin 2 localization

To explore how Lamin-A establishes differences in the actin meshwork between inner and outer cells, we investigated the levels and localization of key actin nucleators, which revealed a function for Formin 2. Although this formin is widely implicated in actin nucleation during meiosis[49,50], its role during preimplantation development remains unknown[32]. Immunofluorescence for endogenous Formin 2 shows that during the 2- to 8-cell stage it localizes both within the cell nucleus and throughout the cytoplasm (Fig. 5a). However, following inner-outer cell segregation the nuclear-to-cytoplasmic ratio of Formin 2 decreases in inner cells (Fig. 5a). This is consistent with the changes

**Fig. 4 | Lamin-A regulates the F-actin meshwork, which serves as a scaffold to stabilize cytoplasmic Amot. a**, Amot levels are increased in the cytoplasm of inner cells. Immunofluorescence images of Amot in fixed mouse embryos with standardized Amot intensity, $n = 18$ for outer, $n = 5$ for inner **P = 0.0011, Mann-Whitney U test. **b**, Knockdown of Lamin-A by siRNA causes an increase in cytoplasmic levels of Amot in outer cells. siRNAs were microinjected at the 1-cell stage. $n = 18$ for control, $n = 10$ for Lamin-A siRNA **P = 0.003, Mann-Whitney U test. **c**, F-actin meshwork density increases in the cytoplasm of inner cells at the 16-cell stage and in the ICM at the blastocyst stage. Insets show detail of cytoplasmic meshwork (cyt) and absence of actin in the nucleus (nuc). $n = 5$ for 16-cell inner, $n = 6$ for 16-cell outer, $n = 8$ for blastocyst ICM, $n = 25$ for blastocyst trophectoderm **P = 0.0087, ***P = 0.001, Mann-Whitney U test. **d**, Latrunculin A causes Amot disruption. Note the decrease in cytoplasmic Amot in the inner cell cytoplasm. $n = 9$ for control, $n = 10$ Latrunculin A, **P = 0.0061 Mann-Whitney U test. **e**, Knockdown of Lamin-A by siRNA causes an increase in cytoplasmic F-actin levels. siRNA was microinjected into a single cell of the 2-cell embryo. H2B-GFP labels injected cells. Inset shows detailed view of cytoplasmic F-actin. $n = 12$ for control, $n = 10$ for Lamin-A siRNA. ****P < 0.0001, Mann-Whitney U test. **f, g**, Human blastocysts immunostained for Amot (**f**) and stained with phalloidin (**g**) show increased levels of cytoplasmic Amot and F-actin in the ICM compared to trophectoderm (TE). For Amot (**f**), $n = 2$ human embryos (no statistical comparison performed). For phalloidin (**g**), $n = 3$ human embryos. $P = 0.0025$ by paired two-tailed t-test. For human embryos, dots represent individual cells. Fluorescence intensities represent results standardized to DAPI. Bars in dot plots represent median and interquartile range. All statistical tests are two-tailed. Scale bars, 10 μm. Source data are provided as a Source Data file.

we found in the cytoplasmic F-actin meshwork in the mouse and human embryo (Fig. 4c, g) and suggests a mechanism whereby Lamin-A regulates the F-actin meshwork and cell fate via Formin 2 nuclear-to-cytoplasmic localization.

To test whether Formin 2 localization is Lamin-A dependent, we microinjected Lamin-A siRNAs and assessed endogenous Formin 2 localization by immunofluorescence. Lamin-A knockdown causes an increase in Formin 2 levels throughout the cytoplasm of outer cells, which adopt a pattern similar to control inner cells (Fig. 5b). Treatment with leptomycin B, an inhibitor of nuclear export has also been shown to affect nuclear Yap shuttling via changes in the nuclear pore complex in cell culture[18]. In the embryo, we found similar results whereby leptomycin B treatment prevents the increases in cytoplasmic Formin 2 observed in inner cells of control embryos (Supplementary Fig. 6a). We then examined the effects of inhibiting formin function, using a time-restricted treatment of 16-cell embryos with the well-established small inhibitor of formin FH domains SMIFH2. This treatment causes inner cells to display a sparse cytoplasmic actin meshwork and consistent with this, also exhibit higher nuclear Yap localization (Fig. 5c). Conversely, microinjection of 2-cell embryos with a high concentration of mRNA coding for Formin 2 fused to Emerald (Emerald-Formin 2) produces an opposite effect, with outer cells displaying an enrichment in both cytoplasmic F-actin and Amot, and lower nuclear Yap levels (Fig. 5d, e).

The Arp2/3 complex is also a key regulator of actin nucleation in the mammalian oocyte[51]. We tested several antibodies and fluorescent fusion proteins but did not obtain a specific localization pattern for Arp2/3 complex components in the embryo. However, treatment with the Arp2/3 inhibitor CK-666 produces a decrease in cytoplasmic F-actin, as well as changes in Yap localization similar to SMIFH2 treatment (Supplementary Fig. 6b). Conversely, expression of the conserved VCA domain of WASP proteins that activates Arp2/3-mediated actin nucleation[52,53] increases F-actin and Amot levels in the cytoplasm of outer cells, and decreases Yap nuclear localization (Supplementary Fig. 6c). These results unveil a mechanism whereby decreasing Lamin-A L:N with cell internalization triggers a greater cytoplasmic fraction of Formin 2, and likely also Arp2/3, regulating cell fate via the F-actin meshwork.

## Discussion

Our data reveal that mechanical forces not only determine inner-outer cell position in the preimplantation embryo, but also elicit changes in actin organization via the nuclear lamina to regulate Yap and Cdx2, ultimately specifying trophectoderm versus ICM identity (Fig. 6). Recent studies in cell culture show that the nucleus can detect mechanical stimuli[54,55]. This may enable cells to sense changes in their shape and position within complex three-dimensional tissues. Yet, the inaccessibility of intact mammalian tissues has precluded dissection of mechanisms linking mechanical forces to cell fate under physiologic conditions. Here, we followed inner-outer cell segregation during the 8- to 16-cell stage, which is driven by differences in actomyosin

contractility[2]. Contractile forces are higher at the apical cortex, where myosin II accumulates. Our results indicate that these forces are relayed to the nuclear lamina via the F-actin meshwork, which provides a continuous contractile system linking the cortex and cytoplasm. Consistently, as cells internalize and gradually lose their apical region, they display lower Lamin-A L:N compared to outer cells, which maintain an apical cortex and high levels of phosphorylated myosin II. F-actin structures serve as mechanical tethers between the cortex and nucleus in other systems, such as the actin cables in *Drosophila* nurse cells[56]. It is thus plausible that the F-actin meshwork exerts forces on the nuclear lamina, which could include both pulling or pushing on lamina components. Yet, we cannot experimentally quantify forces transmitted along this meshwork in the intact embryo. Nevertheless, the idea of mechanical tethers linking cortex and nucleus is supported by our results using azido-blebbistatin to reduce contractility, and SUN1/2 knockdown to decouple the lamina from the cytoskeleton. Cortical forces may also trigger molecular signaling cascades, which could include cytosolic phospholipases[54,55].

Our results show that a downstream effect of Lamin-A's response to mechanical forces is the regulation of Yap shuttling and expression of Cdx2, one of the key transcription factors specifying trophectoderm cell identity. Whereas physical forces have been demonstrated to alter Yap[57], the mechanisms that could link forces to Yap in the embryo remain unclear[58]. Current models are centered on the role of cell polarity, based on differences in apical versus basolateral domains of the inner and outer cell cortices[9,12,58,59]. These models have been supported by pharmacologic and gene knockout manipulations of Rho GTPases. For example, Cdc42 knockout mice fail to implant due to disrupted Cdx2 expression[60]. Similarly, treatment with Rho GTPase inhibitors such as Y27632[61] increase cytoplasmic Yap in outer cells. However, Rho GTPase manipulations can not only affect cell polarity, but also cortical tension and F-actin levels[62]. This makes it difficult to isolate the specific role of polarity in lineage specification. Therefore, our data support an additional mechanism that precedes the establishment of differences in polarity between cells, whereby mechanical forces driving inner-outer segregation can regulate Yap (Fig. 6).

Amot is central to the regulation of Yap phosphorylation in the embryo. Based on its enrichment at the apical cortex, it has also been proposed to be primarily regulated by differences in cortical polarity[9,12,58,59]. Whereas a cytoplasmic Amot fraction has been detected[7], neither the dynamics nor function of this Amot pool have been explored. Here, we imaged Amot across a broad dynamic range of fluorescence intensities, which enabled us to explore differences in its cytoplasmic fraction and found a significant cytoplasmic upregulation in inner cells. Similarly, when staining for F-actin, its high concentration at the cortex dominates over the cytoplasmic fraction. As a result, previous studies have largely focused on the function of cortical F-actin. Imaging across a broader dynamic range in freshly prepared embryos also enabled us to resolve an extensive F-actin meshwork throughout the cytoplasm. Importantly, this meshwork is differentially

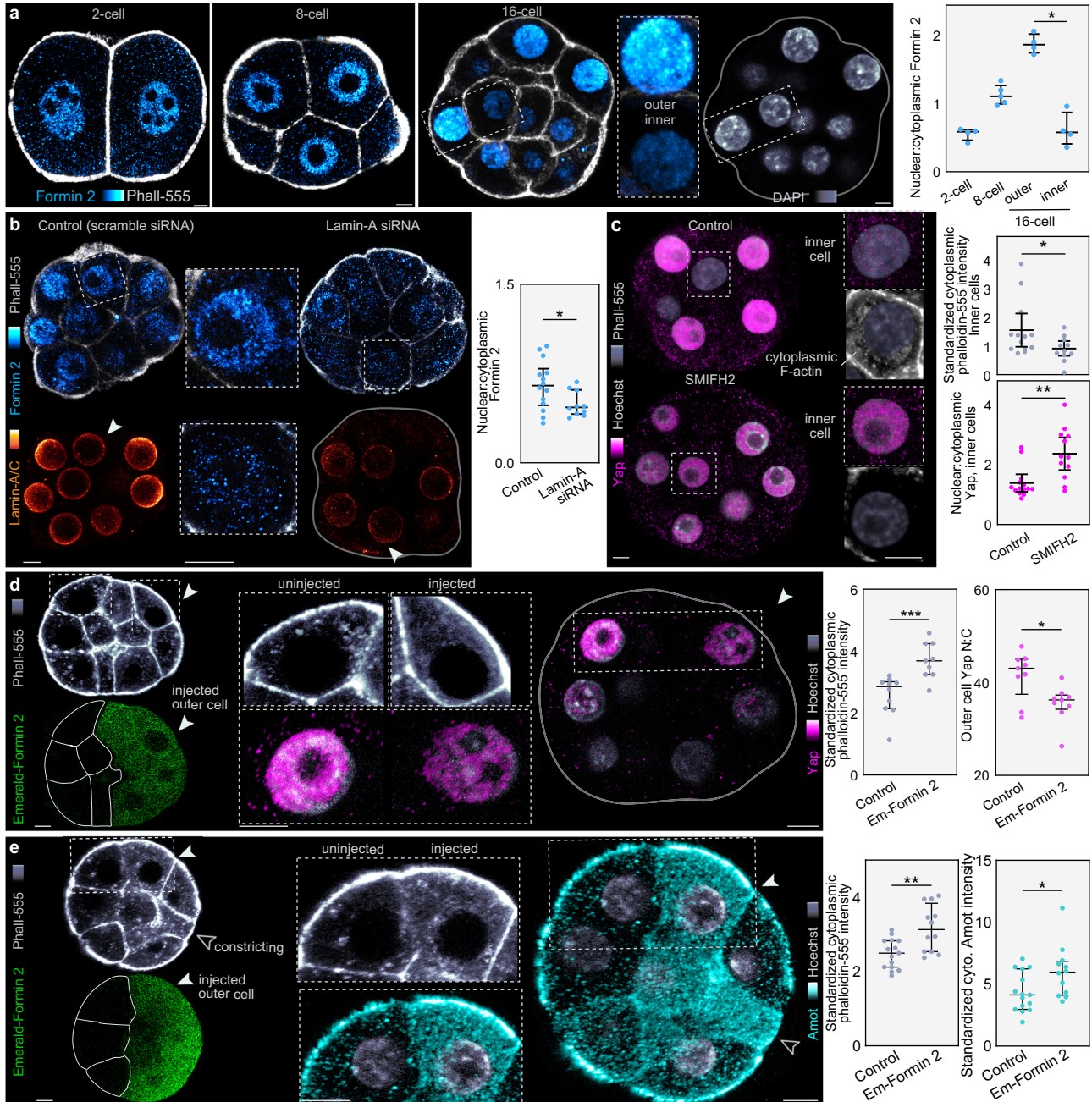

**Fig. 5 | Lamin-A regulates fate by controlling actin nucleator localization.**
**a**, Formin 2 shows a high nuclear:cytoplasmic ratio, which is increased in outer cells following cell internalization. $n = 4$ for all stages. *P = 0.0286, Mann-Whitney U test. **b**, Knockdown of Lamin-A by siRNA reduces nuclear:cytoplasmic Formin 2. siRNA was microinjected at the 1-cell stage. Insets show reduced levels of Lamin-A in siRNA. $n = 14$ for control, $n = 10$ for Lamin-siRNA. *P = 0.0470, Mann-Whitney U test. **c**, Treatment with SMIFH2 reduces inner cell F-actin and increases nuclear:cytoplasmic Yap. 16-cell embryos were either treated with DMSO (control) or 250 μM SMIFH2 and immunostained for Yap and with phalloidin. Insets highlight the differences between inner cell cytoplasmic phalloidin intensity and nuclear Yap intensity between control and treated embryos. *P = 0.0364, **P = 0.0024, Mann-Whitney U test. **d**, **e**, Emerald-Formin 2 overexpression. Emerald-Formin 2 RNA was microinjected into a single cell of the 2-cell embryo, fixed at the 16-cell stage and immunostained for either Yap or Amot. Injected cells can be identified by Emerald-Formin 2 expression (arrowheads). Injected outer cells show increases in cytoplasmic phalloidin, nuclear Yap and cytoplasmic Amot intensities compared to uninjected cells. For panel (**d**) $n = 10$ for Emerald-Formin 2 (injected cells), $n = 9$ for control (uninjected cells). ***P = 0.0006, *P = 0.0244, Mann-Whitney U test. For panel (**e**) $n = 9$ for control (uninjected cells), $n = 10$ for Emerald-Formin 2 (injected cells). **P = 0.0063, *P = 0.0414, Mann-Whitney U test. Fluorescence intensities represent results standardized to DAPI. Bars in dot plots represent median and interquartile range. All statistical tests are two-tailed. Scale bars, 10 μm. Source data are provided as a Source Data file.

organized between inner and outer cells in a Lamin-A–dependent manner and serves as a scaffold for Amot stabilization. This is demonstrated by Lamin-A knockdown, resulting in both increased cytoplasmic F-actin and Amot levels in outer cells. In addition, FRAP results show a larger immobile fraction of Amot throughout the cytoplasm of inner cells, and F-actin disruption with Latrunculin A decreases cytoplasmic Amot levels. Thus, these findings expose a plausible mechanism by which mechanical forces differentially regulate actin organization and Amot stabilization during inner-outer segregation to influence Hippo pathway components specifying fate.

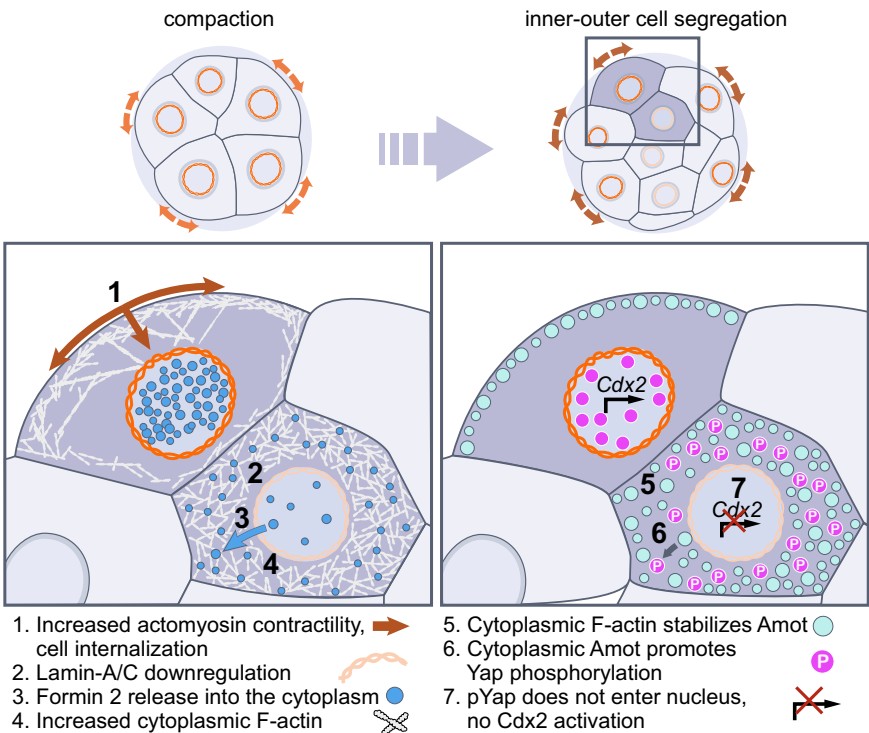

**Fig. 6 | Schematic summary of main results.** Schematic summary shows the main events allowing Lamin-A to link changes in mechanical forces to cell fate in the embryo.

Lamin-A establishes differences in cytoplasmic F-actin via the localization of actin nucleators. We found that Formin 2 is localized in both, the cytoplasm and nucleus and this ratio depends on developmental stage and inner-outer position. Furthermore, its nuclear-to-cytoplasmic ratio can control transcriptional regulators specifying cell fate. This is shown with Formin 2 inhibition, which causes higher inner cell nuclear Yap levels, and with Formin 2 overexpression, which increases cytoplasmic F-actin and Amot, and decreases nuclear Yap in outer cells. The nuclear-cytoplasmic distribution of Formin 2 is dependent on Lamin-A levels, yet the precise mechanism underlying how Lamins control nuclear transport remains unknown[63]. Nevertheless, these findings are in line with recent biophysical manipulations of cultured cells identifying a mechanism of nuclear mechanotransduction dependent on changes in nuclear pore complex conformation[64]. Thus, it is possible that the changes in the nuclear lamina as cells internalize may similarly regulate nuclear transport. Here, we show that both leptomycin B treatment and Lamin-A knockdown decreases cytoplasmic Formin 2 levels, supporting a role for nuclear export. The relationship between cell position in the embryo with nuclear stretch and import/export will be future areas of investigation. In addition to Formin 2, our data suggest a similar role for the Arp2/3 complex, another major F-actin nucleator[53]. Although we could not identify antibodies providing a specific signal for members of the endogenous Arp2/3 complex in the embryo, VCA expression, which initiates Arp2/3-mediated actin branching, results in a phenotype similar to Formin 2 overexpression. Studies in other systems have also reported a role for actin nucleators within the cell nucleus[65]. Thus, it will be of interest to explore if Formin 2 and Arp2/3 could also regulate lineage specification directly inside the cell nucleus.

Markers distinguishing totipotent blastomeres from pluripotent and trophectoderm lineages are highly valuable for reprogramming strategies[66]. Substantial progress has been made in characterizing genetic and epigenetic markers of this developmental divergence. We found that differences in actin organization can also distinguish early blastomeres from trophectoderm and pluripotent ICM. Future studies may examine possible links between the F-actin meshwork and transcription factors associated with pluripotency. Furthermore, it has been shown that following internalization, inner cells do not re-locate to an outer position[2,14]. Thus, the different actin organizations reported here may also contribute to maintaining the first spatial segregation of inner and outer cells during development. This is consistent with studies showing that early cell lineages spatially segregate with cells displaying similar mechanical properties[67–69]. The dense cytoplasmic F-actin meshwork of inner cells may also prevent accidental cortical polarization[11–13] to ensure the fidelity of the ICM-trophectoderm segregation. Finally, the scarcity of human embryos for research has limited our knowledge of lineage segregation to the mouse embryo or stem cell-based, embryo-like structures[70–72]. Our high-resolution images of human embryos demonstrate that differences in endogenous Lamin-A/C L:N also distinguish ICM and trophectoderm lineages and respond to changes in cortical tension. Additionally, we identified abundant cytoplasmic F-actin and Amot in the ICM. This indicates a conserved mechanism of cortical forces leading to Lamin-A–dependent regulation of Yap and Cdx2 in human. As these events occur during human embryo culture, future systems could support physiologic mechanosensing processes to facilitate proper lineage specification.

## Methods

### Ethical approval
This research complies with all relevant ethical regulations, including review and approval of all mouse work from the Institutional Animal Care and Use Committee (IACUC) of the University of Pennsylvania (Protocol #806983) and review and approval of donated human embryo work by the New England Institutional Review Board (WO 1-6450-1).

### Mouse embryo work
Superovulated wild-type female mice at 8 weeks of age were used following animal ethics guidelines of the University of Pennsylvania IACUC. Approximately 5 female mice were used per experimental

replicate. Superovulation was performed using 5 international units (IU) of pregnant mare serum (PMS, National Hormone and Peptide Program) gonadotropin given intraperitoneally and 5 IU of recombinant chorionic gonadotrophin (CG, National Hormone and Peptide Program) given 48 hours after and immediately before mating. The embryos used in our experiments were derived from 4 different mouse strains with distinct genetic backgrounds. These strains included 2 inbred (FVB/NTac and C57BL/6JInv) and 2 outbred (Hsd:NSA(CF-1) and IcrTac:ICR) animal lines purchased from approved vendors (i.e., Charles Rivers Laboratories, Envigo and Jackson Laboratory). All mice were maintained within a BSL2 animal facility at the University of Pennsylvania in pathogen-free conditions with access to water and food ad libitum, a 12-hour dark / 12-hour light cycle between 07:00 and 19:00 in a temperature ($68\,°F – 76\,°F$) and humidity ($30\% – 70\%$) controlled room.

Embryos were flushed from oviducts with M2 medium (Merck) and cultured in KSOM + AA (Merck) covered by mineral oil (Sigma) at $37\,°C$ and 5% $CO_2$. Live embryos were microinjected with 0.1 to 0.3 pL RNA in injection buffer (5 mM Tris, 5 mM NaCl, 0.1 mM EDTA) using a FemtoJet (Eppendorf). For live-imaging, embryos were cultured in LabTek chambers (Nunc) at $37\,°C$ and 5% $CO_2$ in an incubator adapted for the microscope system (Carl Zeiss LSM780, LSM880 and Leica SP8). DNA constructs were cloned into pCS2+ vector for mRNA production. The mMESSAGE mMACHINE® SP6 kit (Ambion) was used to synthesize RNA using linearized plasmids as templates following manufacturer's instructions. RNA was purified using RNAeasy kit (Qiagen) following manufacturer's instructions. Embryos were microinjected with mRNA for: mRuby-Lamin-A at 50 ng $\mu l^{-1}$; GFP-Utrophin at 70 ng $\mu l^{-1}$; memb-GFP at 70 ng $\mu l^{-1}$; RhoA Q63L at 50 ng $\mu l^{-1}$; H2B-RFP at 5 ng $\mu l^{-1}$; H2B-GFP at 5 ng $\mu l^{-1}$; YAP-GFP at 50 ng $\mu l^{-1}$; GFP-MAP2c at 70 ng $\mu l^{-1}$; Emerald-Amot at 100 ng $\mu l^{-1}$; Emerald-Formin 2 at 100 ng $\mu l^{-1}$; Keratin 8-Emerald at 150 ng $\mu l^{-1}$; Keratin 18-Emerald at 150 ng $\mu l^{-1}$; Lamin-A-S22A-Emerald at 150 ng $\mu l^{-1}$; Lamin-A-S22E-Emerald at 150 ng $\mu l^{-1}$; VCA-mRuby at 100 ng $\mu l^{-1}$. siRNAs (QIAGEN) were microinjected at 200 nM. We process a separate set of 5-10 embryos for Lamin-A staining when we start an experiment assessing the effect of Lamin-A downregulation on each new marker to further confirm the efficacy of the knockdown. Embryo sex was not considered in this study due to developmental stage.

siRNAs used:
Lamin-A:
Mm_Lmna_5: AACAGGCTACAGACGCTGAAG (SI02655450),
Mm_Lmna_6: AAGGACCTCGAGGCTCTTCTC (SI02655457),
Ctrl_Lmna_1: AACTGGACTTCCAGAAGAACA (SI03650332).
SUN1: Mm_Unc84a_1: GACCTTAAAGGTGGAAATAAA (SI01462363),
Mm_Unc84a_2: AAGTCGAGGTTTCCTATATTA (SI01462370),
Mm_Unc84a_3: TGGAGATATTTCAAATATTA (SI01462377).
SUN2: Mm_Rik_1: CCGGTTAGTGTTCGGGTGAAA (SI00912751),
Mm_Rik_2: CACGTAGAACTCCCTGCATAA (SI00912758),
Mm_Rik_3: CAGGTGTATATATGTAGCATA (SI00912765),
Mm_Rik_4: CAGGATTGGAATGGTGGATT (SI00912772).

AllStars negative control: sequence undisclosed by Qiagen (SI03650318)

Embryos showing signs of abnormal or arrested development (∼15%) were excluded following previous criteria (Fierro-Gonzalez et al., 2013; Kaur et al., 2013; Morris et al., 2010). We also previously demonstrated that embryos injected and imaged using similar conditions can generate viable offspring following transfer to pseudopregnant mice (Kaur et al., 2013). For drug treatments, all drugs were diluted in KSOM to the following concentrations: H-1152 (Tocris, 2414) at 50 µM; nocodazole (Sigma, M1404) at 10 µM; azido-blebbistatin (Opto-Pharma, DR-A-081) at 10 µM; SMIFH2 (Sigma, S4826) at 250 mM, CK-666 (Sigma, SML0006) at 250 mM, leptomycin B (Sigma, L2913) at 100 nM. Drugs were applied for 4 hours before embryo fixation.

## Immunofluorescence

For immunolabeling, embryos were fixed in 4% paraformaldehyde in PBS for 30 m at room temperature, washed in PBS containing 0.1% Triton X-100 (PBS-T), permeabilized for 30 m in PBS containing 0.5% Triton X-100, incubated in blocking solution (10% fetal bovine serum in PBS) for 2 h, incubated with antibodies for: Lamin-A (Abcam, ab26300) at 1:200; Lamin-A/C (Santa Cruz, sc-376248) at 1:200; Lamin-B1 (Protein Tech, 66095-1-Ig) at 1:200; Phospho-Lamin-A/C (Ser22) (Cell Signaling, 2026) at 1:200; phospho-myosin II (3671 P, Cell Signaling) at 1:200; Phospho-Yap (Cell Signaling13008) at 1:200; Yap (Cell Signaling, 8418 S) at 1:200; Cdx2 (Abcam, 88129) at 1:200; Amot (gift from H. Sasaki) at 1:200; α-tubulin (Sigma, T6199) at 1:500; Formin 2 (Invitrogen, PA5-65632); Keratin 8 (DSHB, TROMA-I) at 1:20; in blocking solution overnight at 4 °C, afterwards embryos were washed in PBS-T, then incubated with Alexa-Fluor conjugated secondary antibodies (Invitrogen, A32731, A32723, A32733, A32728 and A11006) in blocking solution (1:500) for 2 h and washed in PBS-T. To label F-actin, fixed embryos were incubated with Phalloidin-Alexa Fluor 555 (Invitrogen, A34055), at 1:500, Phalloidin-Rhodamine (Molecular Probes, R415) at 1:500 or SPY555-actin (Spirochrome, SC202) at 1:1000. To label nuclei, fixed embryos were incubated with DAPI (Sigma, 10236276001) at 1:1000 or NucBlue-Hoechst 33342 (Invitrogen, R37605) using 2 drops per 1 ml.

## Live embryo imaging

Live embryos were imaged using a laser scanning confocal (Leica SP8 or Nikon A1RHD25) with water Apochromat 40×1.1 NA objective. Embryos were scanned every 5 to 10 min for long-term imaging. Higher temporal resolution of around 2 minutes were used when imaging cleavage divisions. FRAP was performed at 3.5-times zoom on a 5 µm × 10 µm region of interest, photobleached using the 405 nm laser at 100%, with a pixel dwell time of 6 µs. Azido-blebbistatin was activated locally by illuminating a ROI (3 mm×5 mm) with a 405 nm laser at 20% laser power for 60 s.

## Image analysis

3D visualizations and analysis of embryos were performed using Imaris 8.2 or Imaris 9.7 software (Bitplane). The manual surface rendering module was used for cell segmentation, using phalloidin staining to define the boundary of each cell. When quantification of the cytoplasm and cell cortex were required the segmented surfaces were converted into a binary mask of pixels contained with the surface. Minimum and maximum filters were used to produce two additional masks approximately 1 µM larger and 1 µM smaller than the original. The smaller mask was used to define the cytoplasm and a mask of the cortex was produced by subtracting the smaller of these masks from the larger. Quantification of immunofluorescence intensity and expression levels were performed in Imaris and ImageJ by measuring mean fluorescence intensity at a ROI and standardized to DAPI or Hoechst. Inner cells were identified by segmenting embryos and confirming that no part of the cell surface is in contact with the outer surface of the embryo.

## FRAP

For analysis of FRAP experiments, mean fluorescence intensity at the photobleached region of interest (ROI) was corrected for background fluorescence and normalized to a non-photobleached reference. The average of the pre-bleach fluorescence intensities was set to 100%, and the fluorescence intensity immediately after photobleaching was set to 0%. The normalized mean fluorescence intensities were then fitted with an exponential function. The immobile fraction was calculated by taking $1 - I\infty$, in which $I\infty$ is the normalized mean fluorescence intensity when the intensity recovers to a plateau. All fittings were performed in MATLAB (Version R2018a), and FRAP kymographs were created using the montage tool in Fiji.

## Geometric analysis

For measurements of geometrical parameters sets of points from Imaris 3D segmentations were exported to MATLAB for each cell. To obtain the apical surface area, for each cell the points that are at a distance below a cut off distance with neighboring cells were deleted. The remaining set of points defines the apical surface. Apical surface area was calculated as the sum of the areas of the triangles after using Delaunay triangulation and selecting the free boundary facets. Aspect ratio and Flatness respectively were calculated as follows: $(Lz\text{-Max}(LX, LY)) / (Lz+\text{Max}(LX, LY))$ and $Lz_{api}/Lz$, where Lx, Ly and Lz are the lengths of the sides of the minimal bounding box that encloses the cell, with Lz parallel to the vector defined from the center of mass (CM) of the embryo to the CM of the apical surface. $Lz_{api}$ is the length of the side of the minimal bounding box that fits the apical surface. To measure the distance to CM from each nucleus to the embryo, the CM of each nucleus is obtained from Imaris segmentation statistics as position. For the embryo's CM the membrane or actin channel is used for segmentation to obtain the position. Distance is calculated as the 2-norm of the subtraction vector between each nucleus and whole embryo CM position.

Nuclear deformation index was locally defined as the mean value of the distances between the nuclei surface to the surface of the minimum ellipse that fits the nuclei normalized by the distance from ellipsoid center of mass to the point in the surface. Sets of points from Imaris 3D segmentations were exported to MATLAB for each nucleus. The nuclei were fitted with an ellipse and the distance from each point in the nuclei to the ellipsoid was obtained.

## Statistics and reproducibility

Statistical analyses were performed in Excel and GraphPad Prism. Data were analyzed for normality using a D'Agostino-Pearson omnibus normality test. Variables displaying a normal distribution were analyzed using an unpaired, two-tailed Student's $t$-test for two groups, and ANOVA with Dunnett's multiple comparisons test for more than two groups. Variables that did not follow a normal distribution were analyzed using an unpaired, two-tailed Mann-Whitney $U$-test for two groups, and Kruskal-Wallis test with Dunn's multiple comparisons test for more than two groups. Reproducibility was confirmed by independent experiments.

## Human embryo work

Discarded human blastocysts were donated for research under determinations by the New England Institutional Review Board (WO 1-6450-1). Couples had consented to discard frozen embryos for research by signing a notarized documentation form (see https://www.bostonivf.com/content/editor/Discard-Frozen-Embryos-Consent-New.pdf) and no compensation was attributed for donating discarded embryos to research. All discarded blastocysts were de-identified prior to the thawing process by Boston IVF staff. Discarded embryos were determined as "Not Human Subjects Research". All donated samples in this study were obtained from frozen embryos and culture was terminated before day 14 postfertilization. Previously vitrified embryos were thawed according to the manufacturer's protocol (90137-SO−Vit Kit-Thaw, FUJIFILM Irvine Scientific, USA) and cultured for 1 h in individual drops of 75 μl of Continuous Single Culture Complete (CSC) media with human serum albumin (HSA) (FUJIFILM Irvine Scientific, USA), covered with mineral oil in an incubator at 37 °C, 7% $CO_2$ and 6% $O_2$ until further staining. All human embryo experiments were performed at Boston IVF, Waltham MA and supported by private funding from Boston IVF.

## Reporting summary

Further information on research design is available in the Nature Portfolio Reporting Summary linked to this article.

## Data availability

The source data generated in this study are provided in the Supplementary Information/Source Data file. All other data supporting the findings of this study are available from the corresponding author on reasonable request. Source data are provided with this paper.

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

## Acknowledgements

We thank Peter Lenart for his valuable comments when preparing the manuscript. The mouse work in this study was carried out in the Plachta lab supported by grants from the NIGMS (GM139970-01) and NICHD (HD102013-01A1) to N.P., P50 NICHD (5P50HD068157-10) to N.P. and R.M.S.; and NICHD/ASRM RSDP (K12HD849-36) to R.M.S and Anatomical Society (UK) Doctoral Scholarship to A.A.M. The human embryo work was performed in the embryology lab at Boston IVF supported by independent funding from Boston IVF. We thank the Nikon BioImaging Lab (NBIL, Cambridge MA) for help with embryo imaging experiments.

## Author contributions

R.M.S., A.A.M., Y.A., O.P., P.T., B.H., A.D.-M. and S.B. performed the mouse work. G.A., and D.S. performed the human embryo work. D.S. supervised the human studies and C.D.S. and N.P. supervised the mouse studies. R.M.S. and A.D.-M. designed and created figure schemes. R.M.S. and A.A.M. wrote the paper.

## Competing interests

The authors declare no competing interests.
