## [Peer Review File · Nature Communications]

The nuclear lamina couples mechanical forces to cell fate in the preimplantation embryo via actin organizationEditorial Note: This manuscript has been previously reviewed at another journal that is not operating a transparent peer review scheme. This document only contains reviewer comments and rebuttal letters for versions considered at Nature Communications.

REVIEWERS' COMMENTS

Reviewer #1 (Remarks to the Author):

The authors addressed all my comments and I support the publication of this very interesting manuscript

Reviewer #2 (Remarks to the Author):

The authors have addressed all my comments and I have no further reservations. The manuscript is in good shape for publication.

Reviewer #3 (Remarks to the Author):

The authors have adequately addressed my concerns, and the manuscript has greatly improved. A few minor issues should however still be corrected:

- I understand that the regulation of YAP by Amot has been shown before, but still the authors do not prove that, in their context, it is Amot and not some of the many other yap regulatory pathways that drives the observed response. It is highly plausible, and at this stage I don't think the authors need to do further experiments. However, this caveat should be explicitly acknowledged and the abstract and conclusions toned down accordingly.
- Figure 5c still shows a "ratio of ratios" instead of actual YAP n/c ratios. Please correct to show actual ratios. The proposed explanation for the behaviour of ratios with/without SMIFH2 and inner/outer cells is reasonable, but should be explained in the manuscript and not masked by this "ratio of ratio" quantification.
- Figure 2c: I understand that cell internalization time changes from cell to cell, but the authors could simply compare lamin levels before/after segregation for different cells, even if timings were different. Showing data of one single cell as conclusive is not justified in any case. If obtaining such data is complicated due to challenges in microscopy (which I would fully understand), then these data can simply be removed, and as the authors state the more rigorous analysis of figs. 2a-b should be sufficient to make the point.

After these corrections, the manuscript is in my view ready for publication.

We thank the three experts for their excellent comments, which have helped us to extend and improve our manuscript. We have carefully followed all recommendations and hope that the new data and point-by-point response below address all comments satisfactorily.

Reviewer #1

The authors addressed all my comments and I support the publication of this very interesting manuscript

Reviewer #2

The authors have addressed all my comments and I have no further reservations. The manuscript is in good shape for publication.

We greatly appreciate the favorable assessments from Reviewers 1 and 2.

Reviewer #3

The authors have adequately addressed my concerns, and the manuscript has greatly improved. A few minor issues should however still be corrected:

- I understand that the regulation of YAP by Amot has been shown before, but still the authors do not prove that, in their context, it is Amot and not some of the many other yap regulatory pathways that drives the observed response. It is highly plausible, and at this stage I don't think the authors need to do further experiments. However, this caveat should be explicitly acknowledged and the abstract and conclusions toned down accordingly.

We thank Reviewer 3 for their rigorous dissection of our work. Indeed, there is not an experiment in our manuscript that directly shows the Amot stabilized by increased cytoplasmic F-actin resulting from lower Lamin-A levels in inner cells is responsible for increased Yap phosphorylation. We agree with Reviewer 3 that it is highly plausible that this is the mechanism by which Lamin-A levels regulate Yap phosphorylation due previous studies that establish Amot as a key regulator of Yap during preimplantation development.

However, we feel that this urge for caution is justified and have changed the wording of the abstract and conclusion such that a definitive causal link between Amot stabilization and Yap phosphorylation is not emphasized.

The abstract now reads:

“Low Lamin-A shifts the localization of actin nucleators from nucleus to cytoplasm increasing cytoplasmic F-actin abundance. This results in stabilization of Amot, Yap phosphorylation and acquisition of ICM over trophectoderm fate.”

The discussion now reads:

“Thus, these findings expose a plausible mechanism by which mechanical forces differentially regulate actin organization and Amot stabilization during inner-outer segregation to influence Hippo pathway components specifying fate.”

- Figure 5c still shows a “ratio of ratios” instead of actual YAP n/c ratios. Please correct to show actual ratios. The proposed explanation for the behaviour of ratios with/without SMIFH2 and inner/outer cells is reasonable, but should be explained in the manuscript and not masked by this “ratio of ratio” quantification.

We thank Reviewer 3 for this comment and have added the recommended analysis in **new Fig 5c**. Here we compare the fluorescence intensities of nuclear-to-cytoplasmic Yap within inner cells of control embryos and those treated with the formin inhibitor SMIFH2. Consistent with the observed decrease in cytoplasmic F-actin in treated embryos, we show that inner nuclear-to-cytoplasmic Yap levels decrease (by two-tailed Mann-Whitney test, $P = 0.0024$).

- Figure 2c: I understand that cell internalization time changes from cell to cell, but the authors could simply compare lamin levels before/after segregation for different cells, even if timings were different. Showing data of one single cell as conclusive is not justified in any case. If obtaining such data is complicated due to challenges in microscopy (which I would fully understand), then these data can simply be removed, and as the authors state the more rigorous analysis of figs. 2a-b should be sufficient to make the point.

We thank Reviewer 3 for this comment, we agree that data from single cells is not substantial enough to draw meaningful conclusions from and have therefore removed this data from the manuscript.

After these corrections, the manuscript is in my view ready for publication.